# GRAPH SIGNAL REPRESENTATION OF EEG FOR GRAPH CONVOLUTIONAL NEURAL NETWORK

**Soobeom Jang, Seong-Eun Moon and Jong-Seok Lee**
School of Integrated Technology
Yonsei University
South Korea
{soobeom.jang, se.moon, jong-seok.lee}@yonsei.ac.kr

## ABSTRACT

In this paper, we present an approach for graph signal representation of EEG toward deep learning-based modeling. In order to overcome the low dimensionality and spatial resolution of EEG, our approach divides the EEG signal into multiple frequency bands, builds an intra-band graph for each of them, and merges them with inter-band connectivity to obtain rich graph representation. The signal features on the vertices are also obtained from EEG. Finally, the graph signals are learned with graph convolutional neural networks. Experimental results on visual content identification using EEG are presented and various ways of defining intra-band and inter-band connections are examined.

## 1 INTRODUCTION

Graph signal processing is an attempt to analyze signals having irregular structures (Ortega et al. (2017)), where a signal residing on vertices of a graph (instead of regular intervals or grids) is defined and processed. Along with its development, deep neural networks for graph signals have been also proposed (Defferrard et al. (2016); Kipf & Welling (2016); Pham et al. (2017); Simonovsky & Komodakis (2017); Zhang et al. (2018)).

The brain signal typically has an irregular structure, which can be modeled as a graph signal. Since it contains much information of the human mental state, several applications such as neurological disease detection, emotion recognition, and behavior modeling have been developed. An important aspect of the brain signal is that there is certain relationship between signals from different regions, called connectivity (Bullmore & Sporns (2009)). Physically close regions tend to show similar signal patterns due to the volume conductance effect. Furthermore, activities of different regions are often related, which can be described by correlation, synchronization, coherence, etc (den Heuvel & Pol (2010); Hassan et al. (2014)). These types of connectivity relationship have been modeled with graphs in literature (Fallani et al. (2014)).

There exist recent studies applying graph signal processing to brain signals, e.g., for feature extraction (Huang et al. (2016)) and dimensionality reduction (Rui et al. (2016)). In addition, neural networks for graph signals have been considered for analysis of MEG signals (Guo et al. (2017)) and fMRI signals (Ktena et al. (2017)). However, graph signal-based machine learning approaches for EEG are rarely found. Due to the limited spatial resolution of EEG, it is often not straightforward to represent it with an abundant graph structure.

In order to overcome this limitation, we present an approach to obtain rich graph signal representation of EEG and learn it with neural networks for graph signals. We apply our approach to visual content identification based on EEG and show its effectiveness.

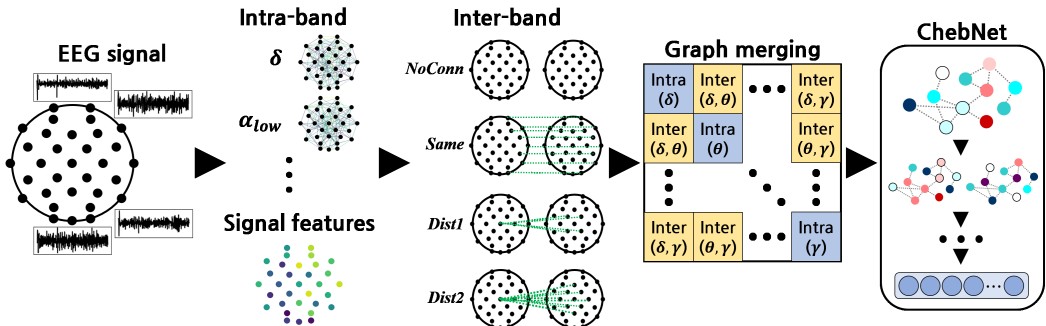

Figure 1: Illustration of our approach.

## 2 METHOD

Our approach is illustrated in Figure 1. A raw EEG signal is acquired from $N = 32$ electrodes. The raw signal with high temporal resolution and wide frequency bandwidth is temporally downsampled and band-pass filtered. This pre-processed EEG signal is decomposed into eight frequency bands, i.e., delta (0-4Hz), theta (4-7Hz), low-alpha (8-10Hz), high-alpha (10-12Hz), low-beta (13-16Hz), mid-beta (17-20Hz), high-beta (21-29Hz), and gamma (30-45Hz). We use the power and entropy of each band signal as the signal features of the graph signals (Koelstra et al. (2012); Sabeti et al. (2009)). To construct a connectivity graph for the whole signal, we first define an intra-band graph for each frequency band and then merge them with inter-band connectivity.

We adopt three methods to define intra-band connectivity: correlation (*Corr*), distance (*Dist*), and random (*Rand*). The correlation connectivity is a type of functional connectivity (Hassan et al. (2014)), where the absolute value of the Pearson correlation coefficient between signals of two electrodes is used as the edge weight between the two electrodes. Second, the distance connectivity is based on the physical distances between electrode positions (Rui et al. (2016)). A Gaussian function is applied to transform the distance to the edge weight: $e^{-d^2/\sigma^2}$, where $\sigma$ is a scaling constant and $d$ is the spatial distance. Therefore, the edge weight has a maximum of 1 and decreases as the spatial distance increases. For the *Corr* and *Dist* intra-band graphs, we keep only the edges corresponding to the largest four weights for each vertex in order to avoid overly connected graphs. Third, we build Erdös-Rényi random graphs with an edge probability of 0.5.

Once intra-band graphs are constructed for all the eight bands, we merge them using three options: no connection (*NoConn*), electrode-wise connection (*Same*), and distance-based connection (*Dist1* and *Dist2*). The first option, *NoConn*, simply collects the intra-band graphs without assigning edges between vertices of different bands. In this case, the adjacency matrix and its Laplacian of the merged graph become block diagonal matrices. The second option, *Same*, assigns connection between the vertices corresponding to the same electrode. Therefore, the adjacency matrix of the merged graph has blocks of identity matrix of size $N$ in the off-diagonal part. The distance-based option (*Dist1* and *Dist2*) is based on the distance connectivity, used for defining intra-band graphs. We keep top-4 (*Dist1*) or top-8 (*Dist2*) edge weights between a vertex in an intra-band graph and vertices in another intra-band graph.

As a result, the EEG signal is converted to a graph signal on $8 \times N = 256$ vertices.

## 3 EXPERIMENTS

We apply the presented method to a classification problem, where the goal is to identify the visual content that the subject watched based on EEG. We employ the DEAP (Koelstra et al. (2012)), which contains EEG signals of 32 participants, recorded during watching 40 sixty-second-long music video stimuli. Each signal has a sampling rate of 128Hz. We divide each signal using a moving window having a length of three seconds and an overlap of two seconds to evaluate short-time identifiability.

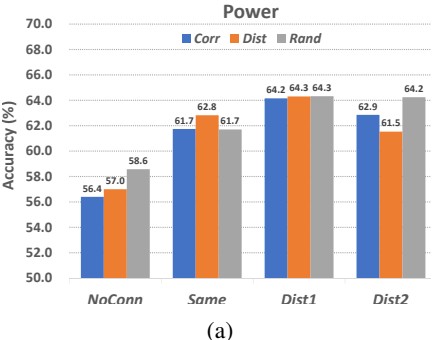
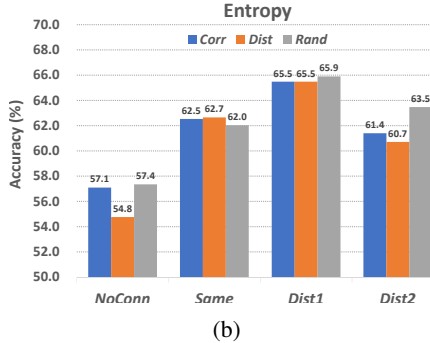

(a)            (b)

Figure 2: Accuracy of EEG-based visual content identification. (a) Power features for signals (b) Entropy features for signals.

We use the graph convolutional neural network, also known as ChebNet (Defferrard et al. (2016)) to classify the graph signals. We convert the connectivity matrix for all frequency bands to its normalized Laplacian, which is fed to ChebNet as input. The employed ChebNet structure has seven layers: two Chebyshev graph convolutional layers with 64 filters of 16th-order polynomials, a graph max-pooling layer of size 2, two Chebyshev graph convolutional layers with 128 filters of 9th-order polynomials, a graph max-pooling layer of size 2, and finally a fully connected layer with 40 output units. We use the cross-entropy loss function and the Adam optimizer (Kingma & Ba (2014)) with L2-regularization for training. The learning rate is set to 0.001 initially and decreases by 0.95 at every epoch. The training epoch is set to 30. We divide the signal segments into training and test sets with a ratio of $80 : 20$.

## 4 RESULTS

Figure 2 summarizes the results. It is observed that graph signal representations of EEG are successfully learned by ChebNet, resulting in accuracies significantly higher than random chance (2.5%). The baseline accuracies by k-nearest neighbor and random forest classifiers are 48.5% and 51.3%, respectively, which are significantly outperformed by our approach. This demonstrates that it is useful to consider graph structures appearing in EEG.

According to the results, an important factor in graph signal representation is how to select an appropriate graph construction method. Especially, the type of inter-band connectivity significantly affects the performance, while the type of intra-band connectivity has only a minor influence. The overall tendency is that a richer connection between different frequency bands yields better performance. Thus, it is important to allow information propagation from a frequency band to another through inter-band connections. However, *Dist2* results in lower accuracy than *Dist1*, showing that excessive connections are not helpful for effective information propagation among regions. *Rand* shows smaller accuracy difference with respect to different inter-band connectivity options than *Corr* and *Dist*, since the graph structure in this case is not related to the EEG patterns. When the two signal features are compared, the entropy feature shows slightly better performance than the power for *Dist1*, but this depends on the other conditions such as intra-band and inter-band connections.

In conclusion, it is beneficial to exploit graph signal representations and, in particular, rich graph structures in order to effectively learn the EEG patterns useful for solving the classification problem. In our future work, our approach will be applied to other types of brain signals and graph signal neural networks.

ACKNOWLEDGMENTS

This work was supported by Basic Science Research Program through the National Research Foundation of Korea (NRF) funded by the Korea government (MSIT) (NRF-2016R1E1A1A01943283).

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
