# OpenReview forum: "Graph Signal Representation of EEG for Graph Convolutional Neural Network"
_ICLR.cc/2018/Workshop — Reject_

### Official Review · AnonReviewer2 · 2018-03-09
**Review: Graph Signal Representation of EEG for Graph Convolutional Neural Network**

**Rating:** 6
**Confidence:** 4

**Review:**

In this paper, the authors present an approach to represent an electroencephalogram (EEG) signal as a graph, capturing within-frequency and cross-frequency interactions. The authors then input this graph representation into a graph CNN (ChebNet) to classify EEG recording segments during which subjects watched different video clips. The approach exhibits good classification accuracy.

This is an interesting problem with nice results, but the presentation is a bit difficult to interpret. Why does the choice of intra-band connectivity appear not to matter for accuracy (Corr, Dist, Rand)? Also, the authors should be careful not to include the DC component in the delta frequency band (generally 0.5-4 Hz).

---

### Official Review · AnonReviewer1 · 2018-03-10
**Graph Conv Network for EEG**

**Rating:** 5
**Confidence:** 5

**Review:**

The authors present a graph signal representation of EEG where the EEG signal is divided into multiple
frequency bands, then an intra-band graph is constructed and merged. This representation is the basis for an EEG classification task.
While the graph representation is interesting, the paper is preliminary from the EEG analysis point of view.

- the data is split 80:20, unclear how. More important is that the EEG typically contains dependencies, trends etc. This requires sophisticated (cross-)validation proceedures, e.g. block-wise cross-validation or leave one subject out CV. So it is unclear whether the result is good or bad.
- No comparison to standard linear EEG classification proceedures for the task is given.
- No physiological interpretation of the result is given, the presented data could be a good classification because of an unphysiological artifact or because it's a great algorithm, unclear at this point.

---

### Official Review · AnonReviewer3 · 2018-03-12
**Review of Graph Signal Representation of EEG for Graph Convolutional Neural Network**

**Rating:** 9
**Confidence:** 3

**Review:**

This paper builds on the recent advances in deep network architecture for processing data represented as graph structures. In the context of EEG, only a few exploratory works have proposed and the authors blame the limited spatial resolution of EEG, that keeps from having rich graph structure.
The authors propose a two-step method to build such structure and evaluate different approaches to do so. The EEG signal is split in F=8 frequency bands. The first step is to define an intra-band connectivity for a given band between the electrode, then to compute an inter-bad connectivity. The EEG signal is thus converted in FxN nodes, where N is the number of electrodes.
The experiments are conducted on DEAP dataset, using ChebNet architecture to process the input graphs. The results are interesting and show that using an inter-band connectivity depending on the distance between electrodes yields the best results, the choice for intra-band connectivity has no specific effect and a random graph offers good performances.
The contribution is very interesting as finding a correct representation of EEG signal is still an important challenge for deep architecture. This paper opens interesting perspectives in the context of ICLR workshop.

---

### Decision · Program_Chairs · 2018-03-20
**ICLR 2018 Workshop Acceptance Decision**

**Decision:**

Reject

**Comment:**

Based on the reviews, this paper has not been accepted for presentation at the ICLR workshop. However, the conversation and updates can continue to appear here on OpenReview. As discussed by one of the reviewers, when encouraging ICLR to study applications such as EEG more, it is important to introduce good experimental practice.